# Synergies and Trade-Offs in the Sustainable Development Goals—The Implications of the Icelandic Tourism Sector

**David Cook** [1,*], **Nína Saviolidis** [2], **Brynhildur Davíðsdóttir** [3], **Lára Jóhannsdóttir** [4] and **Snjólfur Ólafsson** [2]

1    Environment and Natural Resources, School of Engineering and Natural Sciences, University of Iceland, 107 Reykjavik, Iceland
2    School of Business, University of Iceland, 107 Reykjavik, Iceland
3    Environment and Natural Resources, Faculty of Economics and Faculty of Environment and Life Sciences, University of Iceland, 107 Reykjavik, Iceland
4    Environment and Natural Resources, School of Business, University of Iceland, 107 Reykjavik, Iceland
*    Correspondence: dac@hi.is

**Abstract:** The development of major economic sectors can provide the bedrock on which long-lasting national economic prosperity is formed. Iceland's tourism sector is an example of a rapidly expanded industry in recent years, to the extent that it has become the largest sectoral contributor to the nation's economy. The growth of the sector has led to a number of sustainability impacts, thus presenting opportunities and challenges in terms of meeting the 17 Sustainable Development Goals (SDGs) of the United Nations. Using the case study of Iceland, this paper aims to advance the conceptual understanding of the synergies and trade-offs between a nation's tourism sector and performance across the 169 targets of the SDGs. Empirical results were derived from four theme-based focus groups comprised of expert participants, who were tasked with completing scoresheets concerning their perception of the extent of synergies and trade-offs for each target. The majority (126 in number) of the mean scoresheet outcomes for the SDG targets revealed neither synergies nor trade-offs. However, 32 synergies and 11 trade-offs were identified. Many of the target synergies related to new economic opportunities, such as jobs, employment, and training for young people. Target trade-offs tended to be environmental and social. In particular, concern was voiced about the greenhouse gas emissions of the Icelandic tourism sector, which derives from international aviation, cruise ships, and rental car usage. The outcomes of this study are of particular relevance to tourism companies, policy-makers, and governance institutions, all of whom are increasingly endeavouring to link their activities with the fulfilment of the SDGs, maximising synergies, mitigating the extent of any potential trade-offs, and potentially transforming trade-offs into synergies. Furthermore, the results are likely of interest to academics focused on researching the broad sustainability impacts of economic sectors and their contribution to meeting the visionary goals of the SDGs.

**Keywords:** decision-making; tourism; sustainable development goals; Iceland; synergies; trade-offs

## 1. Introduction

Concerns about the sustainability of natural resources and a need for sustainable development have been expressed and reiterated over the years in a series of global political gatherings. These have included Our Common Future in 1987, the Earth Summit of 1992, the World Summit on Social Development in 1995, the World Summit on Sustainable Development in 2002, and Rio + 20 in 2012 [1,2]. The 17 United Nations' Sustainable Development Goals (SDGs) (a schedule of all of the Sustainable

Development Goals and their respective targets are provided in numeric order in Table S1 to this paper) have been widely acclaimed as the culmination of this global dialogue, transitioning from the Millennium Development Goals to provide a comprehensive global blueprint for a route to a more sustainable future and confronting challenges linked to poverty, climate change, inequality, environmental degradation, and securing peace, justice, and prosperity [3].

The 17 SDGs and their respective targets are interconnected, containing synergies but also trade-offs (in this study, we define synergies and trade-offs in accordance with the study by Singh et al. (2018) [4]. Synergies are understood to be co-benefits, occurring in alignment with the various activities of the Icelandic tourism sector. Trade-offs are hindrances and drawbacks linked to the same set of activities) that may be difficult to reconcile [4–7]. This is perhaps most clearly evidenced in relation to Goal 8, 'Decent work and economic growth', which sets a target for all countries to sustain per capita economic growth in accordance with national circumstances [3]. Many economists have argued that maintaining stocks of natural resources should be allocated priority over the flows of income and economic growth sourced from their depletion [8,9]. Such 'strong sustainability' arguments emphasise the limited substitutability of natural for produced forms of capital, and in so doing shift the management objectives of an economy towards the pursuit of a sustainable yield of renewable resources [10–13].

As Hall et al. (2015) articulated, pursuing economic growth entails trade-offs: "Despite repeated attempts to posit sustainable forms of development, including with respect to alternative and sustainable tourism, the global ecological footprint of humanity continues to grow and run down the stock of the world's natural capital. In other words, the achievement of sustainable development via economic growth strategies, even if they constitute so-called green growth, appears extremely difficult if not impossible" [14] (p. 28). National compliance with the overarching growth objective, targets, and indicators of goal 8 may lead to trade-offs relating to goals such as numbers 11, 12, 13, 14, 15, and 16. Equally, synergies may exist between goal 8 and other goals, such as 1, 2, 3, 4, 5, and 6. The extent and character of these trade-offs and synergies are likely to vary given the context of the nation, whether it is a developed or developing economy, and the extent to which a nation's economic expansion is delivered through reliance on the growth of a single industrial or service-based sector. This is evident in the case of the tourism sector, which is a major driver of economic growth in both developing and developed nations [15,16].

Although there has been general academic discussion concerning the potential impacts of tourism activities on the SDGs [17,18], so far no academic study has sought to evaluate the extent of synergies and trade-offs between a national tourism sector and the goals of the 17 SDGs. Thus, this paper's aim is to evaluate the extent to which a national tourism sector stimulates synergies and trade-offs linked to the pursuit of the SDGs, including their respective targets. The selected case study for this task is Iceland, which is the nation with the fastest rate of economic growth in the OECD in recent years, predominantly due to its burgeoning tourism sector [19]. In the period subsequent to the banking collapse of 2008—the largest in history relative to the size of its economy—spiraling bankruptcies and unemployment threatened the sustainability and economic prosperity of the nation [20]. The tourism sector has been the engine of Iceland's economic recovery, with the number of tourists more than quadrupling between 2010–2017, from 488,600 to 2,224,603 [21]. For the first time ever, tourism in Iceland in the period 2013–2017 was responsible for higher foreign exchange earnings (42% in 2017) than exports of marine products (16% in 2017). Over the same time period, the number of people employed in the tourism sector has increased by 68% [21]. The total contribution (direct and indirect) of the tourism sector to GDP amounted to 34.6% in 2017 and this is projected to rise to 40.6% by 2028 [22].

This paper is structured as follows. Section 2 provides a brief literature review of existing publications focused on interactions and trade-offs in the SDGs. Section 3 communicates the recent importance of the tourism sector to the Icelandic economy in terms of growth, and outlines a summary of the known economic, environmental, and social consequences. Section 4 details the methodology for this paper's evaluation, which is based on focus groups and the completion of evaluative scoresheets.

Section 5 combines the results and discussion. It summarizes the results from the focus groups and provides a matrix of the extent to which the Icelandic tourism sector is stimulating synergies and trade-offs across all of the targets of the SDGs. The discussion component focuses on the main implications of the study and provides a broader reflection on the contribution of Iceland's tourism sector towards meeting the SDGs. Section 6 details a brief conclusion and summary of the paper's main implications for policy-makers.

## 2. Overview of Existing SDG Interactions and Trade-Off Studies

Costanza et al. (2016) heralded the publication of the SDGs as "*a global consensus, years in the making*" and "*an important step in the transition to a sustainable world*" [23] (p. 59). The authors also recognized that the publication of the SDGs, however seminal, was only a starting point. They called for future work analyzing how the goals and targets interconnect, especially their synergies and trade-offs, voicing that this quest demands an interdisciplinary contribution from academics, scientists, and policymakers. Several authors have begun to embrace the challenge. In this brief literature review, a summary details the current approaches to evaluating synergies and trade-offs in the SDGs, together with reports that highlight the various institutional challenges relating to their practical implementation.

Nilsson et al. (2016) detailed a conceptual framework, evaluating the extent to which interactions occur between the 17 SDGs, focusing predominantly on the issues of poverty, equality, environmental conservation, and climate change [4]. As an analytical support tool, the authors outlined a seven-point scale of interactions between SDGs. These are rated from +3 (most positive) to −3 (most negative), with four criteria considered in this evaluation being: (1) reversibility of the interaction; (2) bidirectional attributes of the interaction; (3) extent of the impact of the interaction; and (4) certainty of the interaction. Examples cited of the most positive interactions include ending all forms of discrimination against women, which was deemed by [4] to be indivisible from ensuring the full participation of women and their equal opportunities for leadership. At the other end of the scale, a cited example of the most negative interactions is the pursuit of the full protection of nature reserves, which is specifically linked to goals 14 and 15, and has a trade-off with regard to ensuring public access for recreation. Through their approach, [4] emphasised the importance of governance institutions undertaking mutually reinforcing actions ('policy coherence') to minimise trade-offs [4].

The work of Singh et al. (2018) investigated co-benefits and trade-offs between the targets of Goal 14, 'Life Below Water', and other SDG targets [4]. A framework was developed to consider three hierarchical considerations: (1) the compatibility of the relationship (is it a co-benefit, trade-off, or neutral); (2) the contribution of one SDG target for the fulfilment of another; and (3) whether the compatibility of the relationship should be considered to be context-dependent or not. The workshop was split into 16 sessions with contributing experts from the fields of marine science, economics, ocean governance, and social anthropology. Participants were tasked with populating a matrix representing the seven targets of SDG14 versus the targets of the 16 other SDGs. It was found that all of SDG14's targets are related to the other SDGs, with two out of seven targets being particularly significant. These were the increase of economic benefits to Small Island Developing States and least developed countries, the elimination of overfishing, and illegal and destructive fishing practices. As well as highlighting the general contribution of marine environments to sustainable development, the approach of [4] has potential transferability to work analyzing synergies and/or trade-offs concerning other SDGs.

Nerini et al. (2018) conducted a study that was similar in general focus to Singh et al.'s (2018); however, the spotlight of their attention was shone on Goal 7, 'Ensure access to affordable, reliable, sustainable and modern energy for all' [7]. Synergies and trade-offs were characterised between the pursuit of SDG7 and other SDGs. Using an approach of qualitative content analysis and expert consultation, the authors uncovered 143 synergies and 65 trade-offs linked to 143 targets. In particular, the authors specified three human capacity domains in relation to the synergies and trade-offs linked to SDG7. These were (1) realizing aspirations of greater well-being; (2) building physical and social

infrastructures for sustainable development; and (3) achieving the sustainable management of the natural environment. The authors called for better organisation and connectivity of the evidence, enabling actors to work more effectively together to pursue sustainable development [7].

Bowen et al. (2017) considered some of the same governance challenges highlighted by [7] in relation to the simultaneous delivery of multiple SDGs [6]. The authors also highlighted the example of SDG7, and how compliance necessitates the contribution of various actors and agencies, each with its respective stakeholder interests [6]. Furthermore, [7] reflected on how terminology can have different meanings, with understandings of 'affordable' and 'reliable' varying relative to the national context. Such complexities led the authors to outline three major governance challenges that must be addressed in order to ensure the successful implementation of the SDGs. These were as follows: (1) ensuring collective action by creating inclusive decision spaces for stakeholder interaction; (2) embracing inevitable trade-offs through a focus on the principles of equity, justice, and fairness; and (3) guaranteeing that mechanisms exist to hold societal actors to account regarding their decision-making, policy actions, and outcomes [6].

Stafford-Smith et al. (2017) also addressed challenges in the implementation of the SDGs given the inevitability of trade-offs [24]. As the authors noted in accordance with the observations of [23], across the 17 goals, 42 targets address the means of implementation, whereas SDG17 is entirely focused on implementation, but there is no discussion concerning their various interlinkages and interdependencies. As a consequence, the authors are calling for greater attention to be given to interlinkages across three areas: economic sectors; societal actors; and between and among low, medium, and high-income nations. Seven broad recommendations were delineated by the authors to smooth interlinkages in implementation at a national and global level, covering the issues of: (1) finance; (2) technology; (3) capacity building; (4) trade; (5) policy coherence; (6) partnerships; and (7) data, monitoring, and accountability [24].

Overall, there is a growing body of research that is seeking to better understand and quantify, conceptually at least, the various interactions between the SDGs and their respective targets. The use of scoresheets and evaluative matrices has been adopted as a straightforward means of illustrating the extent of synergies and trade-offs, and to act as a starting point in the process of considering how governance institutions could potentially transform the latter into the former. However, such approaches are yet to be adopted in connection with the impacts of important national economic sectors, including tourism.

## 3. Tourism and Sustainability Impacts in Iceland

Iceland is a sparsely populated island in the North Atlantic Ocean with about 350,000 inhabitants. Around 62% of the population resides in the capital area of Reykjavík and Greater Reykjavík, while the rest of the population lives in the lowlands and around the coastline. About 80% of the island is uninhabited; it is characterised by rugged, volcanic, and mountainous areas with several glaciers, one of them being the largest in Europe. In terms of tourist attractions, Iceland has varied landscapes, many of which are relatively short distances from one another and vast wilderness areas, as well as a diverse array of nature-based activities such as horseback riding, river-rafting, hiking, glacier walks, and more [25]. Iceland's tourism is heavily dependent on its natural attractions, as most tourists visit the country to experience its nature [19,21,26].

A recent book chapter [27] and paper [18] outlined the various economic, environmental, and social sustainability impacts of Iceland's expanded tourism sector. In this section, the aim is not to repeat the level of detail contained in a very recent publication, but rather to provide a succinct summary of the synergies and trade-offs described in its contents. Table 1 summarizes the economic, environmental, and social impacts of relevance to the sustainability of the Icelandic tourism sector. Specific examples are added in the results section based on the observations reported in the focus groups, along with empirical evidence from relevant reports and academic publications. Key synergies and trade-offs reported by [27] relate to Iceland's macro-economy and environment. Although tourism

has contributed to employment and a growing share of gross domestic product, and now constitutes the largest economic sector in Iceland's economy, it has imposed upward pressure on the Icelandic krona, ensuring that it is expensive to live in and visit the nation [28]. Equally, since much of Iceland's tourism is nature-based and the tourists are motivated by a desire to experience the nation's unique landscape features and fragile wilderness areas [26], this creates complexities for governance institutions [27]. There are challenges associated with infrastructure development, including maintaining carrying capacity and crowd management at popular tourist sites, such as the world-renowned locations on the Golden Circle route [27].

**Table 1.** Dimensions of tourism-related synergies and trade-offs in Iceland (structured in accordance with the framework of [29] and informed by [18,27]).

| Type of Impact | Synergies | Trade-Offs |
|---|---|---|
| **Economic dimension** | | |
| Economic environment | Increased expenditure | Localised inflation and national price increases |
| | Creation of employment | Replacement of local with foreign labour |
| | Increase in labour supply | Greater seasonal unemployment |
| | Increased value of real estate | Real estate speculation |
| | Increase in standard of living | Increased income gap between wealthy and poor |
| | Improved investment in infrastructure and services | Opportunity cost of investment in tourism means that other services and sectors do not get support |
| | Increased free trade | Inadequate consideration of alternative investments |
| | Increased foreign investment | Inadequate estimation of infrastructure costs of tourism development |
| | Diversification of economy | Increased free trade |
| | | Loss of local ownership due to increased ownership by investment funds and foreign investors |
| | | Overdependence on tourism for employment and economic development |
| Industry and firm | Increased destination awareness | Acquisition of a poor reputation as a result of inadequate facilities, improper practices or inflated prices |
| | Increased investor knowledge concerning the potential for new competition for investment and commercial activity in the destination | Negative reactions from existing local enterprises due to the possibility of commercial competition |
| | Development of new infrastructure and visitor facilities | |
| | Increase in accessibility | |
| | Improvements in destination image | Inappropriate destination images and brands |
| **Environmental dimension** | Changes in natural processes that enhance environmental values | Changes in natural environmental processes due to air and water pollution, and waste issues |
| | Maintenance of biodiversity | Loss of biodiversity and invasive species |
| | Maintenance and regeneration of habitat and ecosystems | Destruction of habitat and ecosystems exceeding physical carrying capacity |

**Table 1.** *Cont.*

| Type of Impact | Synergies | Trade-Offs |
|---|---|---|
| **Socio-cultural dimension** Community | Strengthening of community values and traditions | Weakening or loss of community values and traditions |
| | Exposure to new ideas through globalisation and transnationalism | Increase in criminal activity |
| | Creation of new community space | Loss of community space |
| | Greater security presence | Social dislocation |
| | Tourism as a general force for peace | Exceeding social carrying capacity |
| | Revival and upkeep of local traditions | Loss of authenticity |
| Psychological/Individual | Increased local pride and community spirit | Tendency towards defensive attitudes concerning host regions |
| | Greater cross-cultural understanding | High possibility of misunderstandings leading to host/visitor hostility |
| | Increased awareness of non-local values and perceptions | Increased alienation due to rapid changes to the local community |

## 4. Research Methods

### 4.1. Focus Groups

This study was based on a series of four focus group interviews with experts, during which participants completed evaluative scoresheets on the extent to which the Icelandic tourism sector is contributing to synergies or trade-offs in meeting the targets of the 17 SDGs. Focus groups were selected as the research methodology for this study due to their capacity to integrate the expertise of relevant experts and use deliberation to stimulate an informed debate [30]. The interactive nature of the debate presented advantages over interviews with individuals, enabling participants to share views, hear the views of others, and perhaps refine opinions in the light of what they have heard [31]. In addition, in contrast to alternative deliberative techniques, such as the Delphi method, focus groups do not seek to arrive at a consensus of opinion, but rather to encompass the full breadth of expert insights, perspectives, and values. Furthermore, focus groups are less resource-intensive than the Delphi method, and do not suffer from the affliction of round-to-round participant dropout [31]. An overview of the method adopted in this study is provided in Figure 1.

In the first step, the SDGs were categorized into four different thematic categories adapted from the Stockholm Resilience Institute [32]. The Stockholm Resilience Centre has grouped the SDGs into three thematic categories: Biosphere (Goals: 6, 13, 14, and 15), Society (Goals: 1, 2, 3, 4, 5, 6, 7, 11, and 16) and Economy (Goals: 8, 9, 10, and 12), with SDG17 as a cross-cutting goal [32]. In this study, the SDGs were grouped according to four categories: Environmental; Economic; Social; and Institutional. Table 2 sets out this study's categorization.

Thus, the categorization diverged from the Stockholm Resilience Centre in the following ways:

- SDG7 on affordable and clean energy was grouped within the economic rather than the social theme because of its emphasis on the affordability of energy.
- SDG16 and SDG17 were placed in the institutional theme in order to facilitate discussion on the cross-cutting issues of institutional capacity and coordination, data collection, and implementation in the context of tourism and the SDGs.

**Table 2.** Categorization of Sustainable Development Goals (SDGs).

| SDG Number | Short Title | Aim |
|:---:|:---:|:---:|
| **Social** | | |
| 1 | No poverty | End poverty in all its forms everywhere |
| 2 | Zero hunger | End hunger, achieve food security, improve nutrition, and promote sustainable agriculture |
| 3 | Good health and well-being | Ensure healthy lives and promote well-being for all at all ages |
| 4 | Quality education | Ensure inclusive and quality education and promote lifelong learning opportunities for all |
| 5 | Gender equality | Achieve gender equality and empower all women and girls |
| 11 | Sustainable cities and communities | Make cities and human settlements inclusive, safe, resilient, and sustainable |
| **Environmental** | | |
| 6 | Clean water and sanitation | Ensure availability and sustainable management of water and sanitation for all |
| 13 | Climate action | Take urgent action to combat climate change and its impacts |
| 14 | Life below water | Conserve and sustainably use the oceans, seas, and marine resources for sustainable development |
| 15 | Life on land | Promote, protect, and restore terrestrial ecosystems, sustainably manage forests, combat desertification, and halt and reverse land degradation, and halt biodiversity loss |
| **Economic** | | |
| 7 | Affordable and clean energy | Ensure access to affordable, reliable, sustainable, and modern energy for all |
| 8 | Decent work and economic growth | Promote sustained, inclusive, and sustainable economic growth, full and productive employment, and decent work for all |
| 9 | Industrial innovation and infrastructure | Build resilient infrastructure, promote inclusive and sustainable industrialisation, and foster innovation |
| 10 | Reduced inequalities | Reduce inequality within and among nations |
| 12 | Responsible consumption and production | Ensure sustainable consumption and production patterns |
| **Institutional** | | |
| 16 | Peace, justice, and strong institutions | Promote peaceful and inclusive societies for sustainable development, provide access to justice for all, and build effective, accountable, and inclusive institutions at all levels |
| 17 | Partnerships for the goals | Strengthen the means of implementation and revitalise the global partnership for sustainable development |

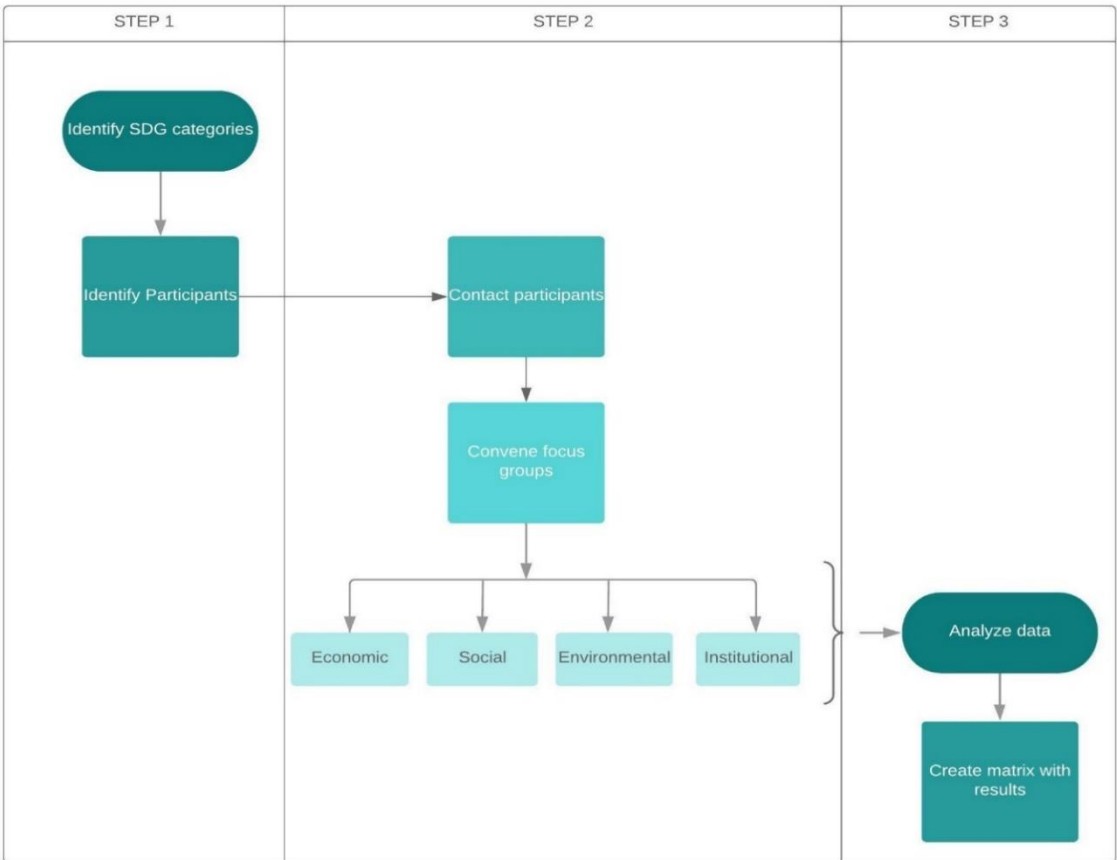

**Figure 1.** Flowchart of the research procedure in this study.

*4.2. Participants*

Once the categories were formed, the researchers identified an initial pool of 53 experts through stakeholder analysis. Close attention was paid to the stakeholder map recently produced in the 'Nordic Tourism Policy Analysis' report [33], which highlighted all the major tourism sector stakeholders in Iceland. Then, expert opinion guided the researchers towards determining and approaching key stakeholders for the theme-based focus groups. The specific participant selection criteria adhered to the approach advocated by [31] and were as follows:

(a)  Purposive sampling: Participants was chosen based on their expected knowledge in terms of the content of each SDG goal, their related targets, and the tourism sector. Participants were contacted by email and informed about the study and its aims. They were also asked to propose an expert to take their place if they were unable or unwilling to participate in the focus groups. This was done to ensure that participants were key informants in their respective fields and to utilise the snowball method.

(b)  Representative sampling: Each focus group had to include participants from various stakeholder groups: business, academia, non-governmental organisations (NGOs), tourism organisations, and governmental institutions.

(c)  Composition: Equal numbers of male and female participants were included in the initial pool of participants to ensure an equitable gender balance.

There were 20 participants in total. Of these, there were eight males (40%) and 12 females (60%). The number of attendees in each thematic focus group was as follows: environmental (6), economic (4), social (5), and institutional (5). The structure of participants was as follows: business (4), academia (5), NGOs (3), tourism organisations (4), and governance institutions (4). With regards

to the participants attending from governance institutions, there were three attendees from national institutes and ministries, and one from the local municipal government in Reykjavik.

The sessions took place between 3–24 April 2019 to test the materials and procedures. The four focus group meetings took place between 10 April–8 May 2019, and each lasted approximately 90 min.

### 4.3. Procedures

Each focus group discussion was moderated by two members of the research team. The moderators' role was to act as observers and facilitators in the discussion and to ensure that all the perspectives were heard and discussed. Materials were distributed in each group with the relevant SDGs and associated targets. Participants were invited to consider and discuss each SDG in their respective thematic group. Each group discussed the extent to which they considered synergies and trade-offs to exist between the Icelandic tourism sector and the targets specific to the SDGs in their respective thematic category. They were also asked to consider how to ameliorate trade-offs through policy-making or other measures. After the focus group, participants had deliberated on each SDG target for which they were asked to score the extent of the trade-off/synergy with the Icelandic tourism sector, with each SDG target evaluated using a seven-point scale. This was the same approach as the one adopted by [5]. The scale was as follows: (−3) strong trade-off; (−2) moderate trade-off; (−1) slight trade-off; (0) neither a trade-off nor a synergy; (+1) slight synergy; (+2) moderate synergy; and (+3) strong synergy (Table S1 to this paper includes all of the evaluative scoresheets used in the four focus groups. For ease of reference, these are arranged in numeric order of the SDGs rather than being grouped according to their thematic categories).

### 4.4. Analysis

The thematic focus group sessions were recorded and transcribed, and participant anonymity was guaranteed. The transcribed data from the discussions was used to enrich the numerical evaluation so as to include lines of reasoning in the final assessment. Each researcher listened to the recordings and summarised them. Then, these summaries were compared to ensure content validity. Finally, all the recorded data will be deleted upon completion of the research project. Results from the scoresheets were averaged and reported to two decimal places for each of the SDGs targets. Then, a straightforward traffic lights system was applied, akin to the indicator evaluation approach of [34], which fed into an evaluative matrix for all of the 169 targets. A red traffic light equated to a trade-off and was linked to a mean score of between −1.00 and −3.00. A yellow traffic was associated with a mean score of between −1.00 and +1.00, meaning that there was neither a synergy nor a trade-off. A green traffic light equated to a synergy and was linked to a mean score of between +1.00 and +3.00.

## 5. Results and Discussion

### 5.1. Summary of Main Outcomes

Table 3 sets out an overall matrix of scoresheet outcomes from the four focus groups. Mean scores (to two decimal places) from participants are provided with respect to each SDG target. Colors for each entry relate to the traffic-lights system of evaluation outlined in Section 4.4 of this paper. The gray spaces reflect cases where a particular target does not exist in relation to a specific SDG. Across the SDGs' 169 targets, there were 32 synergies (18.9%) and 11 trade-offs (6.5%) identified, whilst all the other targets were classed in the neither nor category.

**Table 3.** Evaluative matrix of synergies and trade-offs between Icelandic tourism and SDG targets.

| Target | Sustainable Development Goal | | | | | | | | | | | | | | | | |
|---|---|---|---|---|---|---|---|---|---|---|---|---|---|---|---|---|---|
| | 1 | 2 | 3 | 4 | 5 | 6 | 7 | 8 | 9 | 10 | 11 | 12 | 13 | 14 | 15 | 16 | 17 |
| 1 | −0.20 | 0.00 | 0.00 | 0.00 | −0.40 | 0.00 | −1.25 | 1.75 | 2.00 | 0.25 | −1.00 | 1.25 | 0.80 | −2.20 | −0.20 | −0.80 | 1.00 |
| 2 | 0.60 | −0.60 | 0.00 | 0.00 | −2.40 | 0.80 | −1.00 | 0.00 | 0.00 | 0.50 | 0.00 | 1.75 | 1.00 | −1.80 | 0.00 | −0.40 | 0.00 |
| 3 | 0.00 | 1.00 | −0.20 | 1.40 | 0.00 | −1.80 | 0.25 | 2.25 | 1.25 | 0.50 | 0.00 | 0.75 | 1.20 | −1.20 | −0.60 | 0.80 | 0.00 |
| 4 | 0.00 | 0.40 | 0.00 | 1.80 | −0.20 | 0.00 | | 1.25 | 1.25 | 0.50 | 2.00 | −0.75 | | 0.00 | −0.80 | −0.40 | 0.00 |
| 5 | −0.60 | 0.20 | −0.40 | 0.00 | 1.00 | 0.00 | | 1.25 | 0.50 | 0.75 | 0.00 | 0.75 | | −0.20 | −1.20 | −1.20 | 0.00 |
| 6 | | | 0.20 | 0.00 | 0.00 | 0.00 | | 2.50 | | 0.00 | −0.80 | 0.25 | | 0.00 | 0.00 | 0.00 | 0.00 |
| 7 | | | −0.20 | 1.00 | | | | 0.00 | | 0.75 | 0.00 | 0.25 | | 0.00 | 0.00 | 0.00 | 0.00 |
| 8 | | | 0.00 | | | | | 0.25 | | | | 2.00 | | | −2.20 | 0.00 | 0.00 |
| 9 | | | −0.40 | | | | | 1.75 | | | | | | | 0.00 | 0.00 | 0.00 |
| 10 | | | | | | | | 1.25 | | | | | | | | 0.00 | 0.00 |
| 11 | | | | | | | | | | | | | | | | | 0.00 |
| 12 | | | | | | | | | | | | | | | | | 0.00 |
| 13 | | | | | | | | | | | | | | | | | 0.00 |
| 14 | | | | | | | | | | | | | | | | | 2.00 |
| 15 | | | | | | | | | | | | | | | | | 0.00 |
| 16 | | | | | | | | | | | | | | | | | 1.00 |
| 17 | | | | | | | | | | | | | | | | | 1.60 |
| 18 | | | | | | | | | | | | | | | | | 0.00 |
| 19 | | | | | | | | | | | | | | | | | 0.00 |
| A | −0.80 | −0.60 | 0.00 | 0.00 | 0.20 | 0.60 | 1.25 | 0.25 | 0.25 | 0.00 | 1.40 | 0.50 | 0.00 | 0.40 | −0.20 | 0.00 | |
| B | −0.20 | 0.00 | 0.00 | 0.00 | 0.00 | 1.40 | 1.00 | 0.00 | 1.00 | 0.00 | 1.40 | 0.75 | 0.00 | 0.00 | 0.00 | 0.60 | |
| C | | 0.00 | 0.00 | 0.00 | 0.20 | | | | | 0.50 | 0.00 | 0.25 | | 0.00 | 0.00 | | |
| D | | | 0.20 | | | | | | | | | | | | | | |

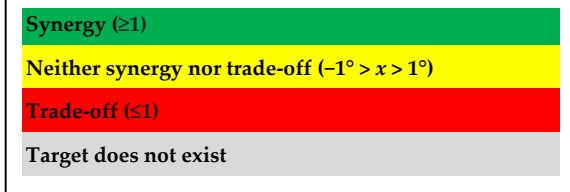

Synergy (≥1)

Neither synergy nor trade-off (−1° > *x* > 1°)

Trade-off (≤1)

Target does not exist

Across six of the 17 SDGs (35.3%), zero synergies were identified. Exactly one-quarter of the 32 target synergies related to SDG8 (decent work and economic growth). Other goals with three or more synergies were SDG4 (inclusive and equitable education), SDG9 (industrial innovation and infrastructure), SDG11 (sustainable cities and communities), SDG12 (sustainable consumption and production), and SDG17 (partnerships for the goals). Out of the 32 target synergies, five had mean outcomes of more than 2.00, equating to moderate to strong synergies. These belonged to SDGs 8 (two targets), 9, 11, 12 and 17. The highest mean outcome across all targets was 2.50, which was identified in connection with SDG8, Target 6 on youth employment.

Trade-offs were identified within seven of the 17 goals (41.2%). However, only SDGs 7 (affordable and clean energy), 14 (life below water) and 15 (life on land) had more than one trade-off, and no SDG had more than the three linked to SDG14. For three of the SDGs with trade-offs—14, 15, and 16 (peace, justice, and strong institutions)—there were no counterbalancing synergies. Out of the 11 trade-offs, three had mean target outcomes of less than −2.00, equating to moderate to strong trade-offs. These were linked to SDGs 5 (gender equality), 14, and 15. The lowest mean outcome and thus the largest trade-offs across all targets was −2.40, which was associated with SDG5, Target 2 (violence against women and human trafficking).

*5.2. Synergies*

5.2.1. Economic

The focus group participants communicated the contribution that Iceland's tourism sector has made to economic growth and job creation, which was reflected in SDG8 having the most target synergies. Two of the targets linked to SDG8 had mean outcomes of more than 2.00, target 3—relating to entrepreneurship, development, and job creation—and target 6 addressing youth employment. Since the collapse of Iceland's banking sector in 2008 [20], tourism in Iceland has been a major driver of economic growth and an aid to economic stability, contributing (both directly and indirectly) about 40–50% of the economic growth in Iceland after 2011 [35]. In 2017, tourism outpaced other sectors in Iceland with 42% in foreign exchange earnings [21], making a direct contribution to gross domestic product (GDP) of 8.6% [36].

During the period 2008–2018, the number of people employed in the tourism sector and related activities grew by 98.5% [37]. Since 2015, there has also been a 40% increase in the number of firms in the Icelandic tourism industry [38]. The contribution of tourism to job creation and economic growth in Iceland appeared to be an underlying factor in the synergies found in relation to targets 1, 3, 5, 6, and 9 of SDG8, with target 9 directly focussed on the topic of sustainable tourism and job creation. Although much of the job creation in Iceland's tourism sector has related to traditional service sector roles, the construction sector has also expanded to try to keep pace with the increased supply of visitors, particularly through the building of hotels and visitor infrastructure [38].

Focus group attendees commented on the contribution that the Startup Tourism initiative has made in stimulating innovation and entrepreneurship across the sector in Iceland, as well as leading to job creation among young persons and economic growth. These were discussed as being central to synergies in SDG8 but also SDG9, which focusses more directly on the subject. The strongest synergy (mean of 2.00) was found in relation to Target 1 of SDG9, addressing the creation of resilient and sustainable infrastructure. Targets 3 (access to credit for developing infrastructure) and 4 (upgrading of infrastructure using clean technologies) of SDG9 were also found to be synergistic. Focus group participants commented on the recent advancements in infrastructure development linked to Iceland's tourism industry, observing the expansion at Keflavík International Airport and the provision of facilities at the most frequented visitor sites, including the Golden Circle. The airport has expanded in size considerably since 2012 to accommodate the increasing numbers of tourists and through-traffic, as it also serves as a hub between Europe and the Americas [35]. There was recognition amongst the participants that the growth of the tourism sector had quelled arguments in Iceland in favour of the

expansion of heavy industries, such as aluminium production, which, although fueled by renewable energy, is carbon intensive.

A total of three synergies linked to SDG12 were reported based on the scoresheet responses of the focus group participants. The strongest of these were associated with targets 2 (sustainable management of natural resources) and 8 (information and awareness about sustainable development). Focus group participants reported that the expanded tourism industry had led to both the need for greater management planning and policy interventions concerning the sustainability of Iceland's natural assets, and in turn had increased awareness of such issues amongst the population. These opinions are reflected to some extent in current government policy, which advocates the adoption of financial incentive instruments in the form of a tourism tax from 2020 onwards [39]. In addition, the government's financial plan for the period 2016–2023 earmarked 2.8 billion ISK to tourism-specific development in protected areas and popular destinations throughout the country [39].

### 5.2.2. Environmental

A total of three synergies were identified by focus group participants in connection with the environmental goals. No synergies were identified with respect to SDGs 14 and 15. One of the target synergies related to the cross-cutting objectives of SDG6 focused on supporting and strengthening the participation of local communities in improving water and sanitation management. Focus group participants opined that local communities around Iceland, whose livelihoods greatly depend on tourism, might envisage more sustainable management of water resources and sanitation as being economically advantageous. As far as the authors are aware, there is no documented evidence showing these effects, particularly in relation to sanitation and water treatment issues. On the contrary, there is anecdotal evidence that some areas have reached capacity limits and may soon need to be upgraded in line with increased use [40]. This is also important in terms of reducing ecological impacts to sensitive areas, such as for example in Lake Mývatn, where inadequate sewage treatment by hotels in the area has threatened the ecosystem [41]. Another report commissioned by the Tourism Task Force assessed access to toilets around the country in 2016. The report found that toilet availability in popular destinations and on the Ring Road that surrounds the island was far from satisfactory and often non-existent [42].

The other two synergies linked to SDG13 (climate action) involved numbers 2 and 3. Respectively, these targets address the integration of climate change measures into national policymaking, and education concerning climate change mitigation and adaptation. With regards to both targets, focus group participants suggested that the Icelandic tourism sector can increase pressure on national and local governments to reduce impacts on the climate, in part due to the importance and image of the sector. The increased adoption of certification schemes for quality and environmental management in Icelandic tourism, such as Vakinn, was cited as an example of the tourism industry leading by example and placing indirect pressure on the national government to enact policies that reduce the impacts of the sector.

### 5.2.3. Social

Across the six SDGS with a social focus, a total of eight synergies were identified, and half of these were linked to SDG4. Synergies were also found to be linked to SDGs 1 and 3.

In association with SDG4, synergies were found in relation to targets 3, 4 and 7. In the case of Target 3, focus group participants expressed an opinion that the Icelandic tourism industry has developed courses and training for people working in the sector. Participants also contended that the Icelandic tourism sector is making a strong, albeit slightly indirect, contribution to education for sustainable development (Target 7), because the national discourse has been focussed on these issues. Although this discourse has not been centered specifically on the term "sustainability", there has always been a lot of discussion about environmental issues such as the soil erosion of footpaths and walkways. In addition, focus group participants discussed the role of tourism in Iceland as a promoter

of peace via the many cultural exchanges that happen when people travel to the nation and return to their homeland with a new perspective.

In relation to the synergy reported for Target 4, focus group participants acknowledged many examples of entrepreneurship in tourism, even in the most remote areas of Iceland, which have led to the creation of jobs for Icelanders and necessitated imported labour. In recent years, Icelandic culture has been broadened through increased immigration, as workers have moved to the country in search of employment within the tourism sector. A recent report on tourism and the labour market in the capital area found that about half of the jobs in the tourism sector have been filled by immigrant workers [43]. Foreign immigration to Iceland has increased by 79% since 2011 [44], with the tourism and construction sectors absorbing most of these workers [28,45]. Many of these workers live and work in new hotels and guesthouses located a considerable distance from the capital city of Reykjavík [43].

Other synergies across the socially themed SDGs were Target 3 of SDG2, Target 5 of SDG5, and Targets A and B of SDG11. With regards to Target 3 of SDG2, multiple focus group participants had voiced the viewpoint that remote rural areas in Iceland appeared to be benefiting from tourism, with local agricultural activities and family farms brought to life again through the emergence of diversified income opportunities. The synergy in Target 5 of SDG5 reflected the observation that women have become more prominent in senior positions across the Icelandic workforce, and, specific to Icelandic tourism, female chief executive officers (CEOs) are in charge of some of the leading companies, including Elding, Icelandair Hotels, and the Radisson hotel chain. The synergy identified in relation to Target A of SDG11 appeared to reflect recognition that the expanded Icelandic tourism sector has stimulated the interest of policymakers concerning how to support the growth of cities and towns around Iceland, and how to ensure a more balanced distribution of visitors across the country. Target B of SDG12 was assessed to be synergistic given that the increased number of people present in Iceland has necessitated greater planning by the relevant authorities on disaster management. This is particularly due to possible evacuations caused by volcanic eruptions or glacial outburst floods, either of which might imperil the ring road around Iceland.

### 5.2.4. Institutional

Four synergies were identified across the two institutionally themed SDGs, all of which related to SDG17. These targets were numbers 1, 14, 16, and 17. With regards to Target 1, focus group participants asserted that the lack of earlier regulation of accommodation platforms, such as Airbnb, has since prompted the tax authorities to clamp down on potential tax evasion practices, albeit they recognized that the practice has not been ameliorated completely.

The strongest target synergy concerned number 14, which had a mean score of 2.00. There was recognition among the focus group participants that the Icelandic tourism sector was playing a strong role in ensuring policy coherence for sustainable development. Comments were made about how the Ministry of Tourism, Industry, and Innovation had formed a Tourism Task Force in 2015, which was required to develop a five-year plan for the sustainable development of the industry. The culmination of this work is currently occurring at the same time as a general national debate about how best to preserve Icelandic nature and develop the tourism industry [27].

Synergies linked to targets 16 and 17 related to partnership building among institutions. Focus group participants communicated that tourism to Iceland was emissions-intensive due to the remoteness of the island and need for most visitors to fly in and out. The Icelandic tourism sector was deemed to be indirectly highlighting the need for international solutions to the problem of greenhouse gas emissions from the aviation sector. In addition, it was stated that the Icelandic and New Zealand governments were cooperating to find common policy solutions to the sustainability challenge of nature-based tourism on a national scale. Domestically, with respect to Target 17, participants acknowledged that municipalities have responsibility for the development and maintenance of Icelandic tourist sites, but receive little or no financial benefit from the tourist flows. Therefore, public–private partnerships have been increasingly adopted to ensure that the supply of infrastructure meets demand.

### 5.3. Trade-Offs

#### 5.3.1. Economic

Trade-offs were identified in only one of the five SDGs with an economic theme. These were targets 1 and 2 of SDG7. In relation to Target 1 on access to affordable, reliable, and modern energy services, focus group participants voiced concerns that access to energy may come at a cost to tourism due to negative effects on the landscape and natural wilderness. The discussion included a debate about competition between Iceland's energy and tourism sectors regarding the value of nature, with the energy sector potentially demanding access to resources that the tourism sector deems valuable enough to prohibit the development of energy infrastructure.

Although the main focus of the focus group debate was on the advantages of energy provision versus the preservation of natural resources for the benefit of tourists, the participants tapped into a wider debate in Iceland about the relative merits of infrastructure provision and what should be prioritised. The pace of tourism growth has outstripped institutional and governmental capacity to respond in a timely fashion, and so various public services and built infrastructure have been put under strain due to the increased numbers of tourists [45]. The airport has expanded in size considerably since 2012 to accommodate increasing numbers of tourists and through-traffic, as it also serves as a hub between Europe and the Americas. Effects of the airport expansion and associated increase in tourist numbers on other infrastructure and services have largely been overlooked [35]. A recent OECD report on Icelandic tourism argued that "major infrastructure decisions . . . need to be based on sound and wide-ranging analysis", taking into account not only economic effects but also social and environmental impacts [28] (p. 34). In part, this gap between policy and infrastructure needs reflects the initial rationale in the policy sphere during the first few years after the economic recession, wherein the tourism industry was conceptualised as one of the production industries in Iceland's economy. As Jóhannesson and Huijbens (2013) put it, "the mentality in regard to tourism development by the central authorities has to a large extent been similar to the production industries where more fish mean more money and larger aluminium smelters mean greater profits" [46] (p. 143).

The trade-off identified in relation to number 2 of SDG7 was in relation to the share of renewable energy in Iceland. Although Iceland is world-leading in this regard, the focus groups nevertheless recognised the negative contribution of the expanded rental car market, given its reliance on fossil fuel combustion. Iceland's transportation system is predominantly based on the private car in terms of the most frequent travel mode within the country. As a result, tourism relies heavily on rental cars, which have increased rapidly in the last few years from around 5000 rental cars in 2006 to 21,000 in 2016 [47]; almost 10% of the car fleet in Iceland is now comprised of rental cars [48]. Apart from the pressures on infrastructure, the increase in cars can lead to more traffic congestion, air pollution [49], and greenhouse gas emissions [50], especially in the capital region. The transportation sector has already been singled out as a major target area for improvement to increase the sustainability of tourism in Iceland [28]; it is also one of the nation's main policy avenues for climate action [51]. This is equally the case with transportation to and from Iceland, which is mostly by air, but there is also a growing volume of cruise ship traffic in the summer months [27].

#### 5.3.2. Environmental

Almost half of all the trade-offs across the SDGs were associated with environmentally themed goals. Three trade-offs were determined in connection with SDG14: two in SDG15 and one in SDG6. Zero trade-offs were identified by the focus group participants in SDG13.

The three trade-offs associated with SDG14 were numbers 1, 2, and 3. All of the concerns voiced by the focus group participants related to the greenhouse gas emissions of the tourism industry in Iceland. In Iceland, greenhouse gas emissions from tourism have been attributed mostly to the transportation sector, with aviation estimated to account for between 50–82% of all tourism emissions, depending on the distance of flights [52]. According to the international bunker fuel data held in relation flights to

and from Iceland, Iceland's emissions from aviation have more than doubled in the period from 2000 to 2016 (the last submission year) [50].

In relation to targets 1 and 2 of SDG14, concerns were also raised about the impacts of cruise ships, with trade-offs discussed concerning their use of heavy fuel oil. Cruise ship tourism has also become a potentially significant source of pollution in the last few years. Cruise ships are associated with a number of negative environmental effects including air pollution, polluting discharges such as sewage, bilge oil, chemicals, and greenhouse gas emissions [53]. These impacts have yet to be quantified in Iceland, although cruise ship passengers have increased from about 28,000 in 2001 to about 145,000 in 2018 [54], which is an approximate increase of 420%. In relation to target 3 of SDG14, the group opined that the greatest threat to ocean ecosystems is acidification, and that this is directly related to the amount of greenhouse gases released. Thus, if tourism in Iceland increases, it will adversely impact the ocean ecosystem, even if indirectly.

A trade-off was also identified in relation to Target 3 of SDG6. The focus group participants were concerned about the impacts of the Icelandic tourism sector on water quality, particularly in small, remote communities. The example of Lake Mývatn was mentioned. Increases in tourism have placed upwards pressure on current facilities, creating the need for upgrades, and focus group participants opined that many very small municipalities are struggling to secure sufficient funds for these.

The trade-off in Target 2 of SDG15 related to concerns about afforestation practices in Iceland. Whether the issues raised were related to tourism is debatable. Participants observed that the trees planted in Iceland are often not native species. The go-to plants for afforestation are often coniferous rather than birch due to their rapid growth. However, when planted in the wrong sites, they can reduce biodiversity; thus they could be deemed to be unsustainable.

The joint-largest trade-off (mean of −2.40) among the environmental goals related to Target 8 of SDG15. This was connected to the potential for tourists to introduce invasive species to Iceland. Focus group participants discussed the potential for freshwater ecosystems to be impacted by alien species through tourism activities, for instance via fishing equipment or wellington boots. Participants also reflected further on the issue of ballast water and cruise ships. According to the group, it makes economic sense for cruise ships to unload ballast water at the ports, since doing this when passengers disembark saves time.

### 5.3.3. Social

Two trade-offs were identified in relation to the socially themed goals. These were Target 2 of SDG5 and Target 1 of SDG11. No trade-offs were found in connection with SDGs 1, 2, 3, and 4.

Target 2 of SDG5 concerned the elimination of all forms of violence against women in the public and private sphere, including human trafficking and sexual exploitation. It was felt that this situation was worsening in Iceland due to the tourism sector. As far as the authors are aware, there are no academic studies that corroborate the opinions of the focus group, although there have been anecdotal reports in the English-language media [55], a critical United States (US) government report on the extent of human trafficking [56], and a recent domestic study by the Icelandic Travel Industry Association on wage exploitation and financial fraud [57].

Target 1 of SDG11 concerns access to safe and affordable housing. Focus group participants raised the issue of immigrant workers in the tourism industry being forced to live in unsuitable accommodation, such as converted garages or industrial buildings. There was also discussion concerning the affordability of housing in Iceland due to a supply shortage spawned by the hosting of tourists within the Airbnb market. Although Airbnb has helped to meet the demand for tourist accommodation, it has also led to fewer available apartments for local residents and increased prices in the housing and rental markets. The Central Bank of Iceland estimates that the number of apartments that were mainly used for short-term lodging through Airbnb were about half to more than two-thirds of the new apartments in 2016 [58]. In total, it has been estimated that 15% of the total rise in real house prices in the period 2014–2016 can be attributed to the growth of Airbnb apartments in that period [58].

Therefore, housing has become less affordable for young people and low-income households [28]. Immigrants in Iceland are particularly vulnerable to increases in prices in the rental market [59]; at the same time it is also more difficult for them to secure rental accommodation [60]. The number of apartments that were used only for short-term renting did not increase in 2018. and although there is still a housing shortage, it is estimated that the supply of housing, especially affordable dwellings, will gradually rise to match demand over the next few years [38].

### 5.3.4. Institutional

A single trade-off was identified, belonging to Target 5 of SDG16. Focus group participants expressed the view that tourism was probably having a countering effect on reducing bribery in all its forms in Iceland. This opinion appeared to be formed from anecdotal evidence about the practices of some tourism companies in Iceland. Examples were cited of hotels selling bottled water to tourists and some restaurants having a tip jar, even though the service charge is included in their menu prices. Equally, the discussion concerning corruption proceeded to focus on issues of rights and power—for example, the individuals and companies who win contracts to provide tourism services, build certain infrastructure, obtain loans, and how these people are connected. Others in the group contended that fixing these issues was not really within the remit of tourism, and these issues were really societal and political challenges for Iceland to address.

### *5.4. Implications of Results*

This paper set out to evaluate the impacts of Icelandic tourism on performance across all of the SDGs and their respective targets, with the aim of determining whether the sector stimulates synergies and/or trade-offs. The majority of the mean outcomes with respect to the SDG targets showed neither synergies nor trade-offs. Overall, this study suggests that the Icelandic tourism sector makes a largely positive contribution towards the meeting of multiple objectives across the SDGs, with evidence of almost three times more synergies than trade-offs. However, several trade-offs pertain to environmental goals, and their incidence and degree should not be understated based on the outcomes from this study.

The significance of Iceland's tourism sector to the national economy was reflected in synergistic effects with SDG8. This was the only SDG to have an overall synergy with the Icelandic tourism sector. This outcome should be of interest to tourism companies in Iceland, employees in the sector, politicians, and agencies seeking to maximise the economic benefits of tourism across the nation, such as the Iceland Tourism Cluster. There is increasing interest around the world in matching company and business sector objectives with the SDGs and their respective targets, and thus one of the main practical advantages of this work is that it identifies, at least specific to Iceland, the links between corporate activities and SDG targets. New entrepreneurial activities linked to tourism in Iceland, aided and abetted by innovative initiatives such as the Iceland Tourism Cluster and Startup Tourism, directly contribute to job creation and synergies with at least five targets in SDG8, especially number 9 on sustainable tourism and job creation.

Businesses specialising in infrastructure works may also wish to take note of the results. Synergies in the environmental sector were identified, including a need for communities around Iceland to have sufficient infrastructure to cope with the influx of tourists. In many cases, built infrastructure has been put under strain in recent years due to the large increase in users over a short time period, and many roads and various facilities, especially in the countryside, are not up to par [40]. The physical condition of the roads is important for tourism in terms of safety and access to certain areas, and can also be an important factor influencing the distribution of visitors around the island. Improved road conditions might reduce the number of incidents that the police and emergency services handle. A prominent example of infrastructural improvement that is important for tourist safety is the changing of single to double-lane bridges, especially on the most frequently used Ring Road around Iceland [40]. The lag in infrastructure development to accommodate the increased numbers of users is partly due to private and public sector oversight, as the soaring popularity of Iceland as a tourist destination was

relatively unanticipated. In some cases, the lack of appropriate infrastructure is related to the lack of tourism revenues for those municipalities that are responsible for development in their regions [28]. The Federation of Icelandic Industries published a report in 2017 on the state and future outlook of built infrastructure in Iceland. The report assessed the current condition of infrastructure and estimated the associated maintenance costs for the coming decade. The assessment found that the road transport system, sewer and drainage systems, and the other airports and landing areas received the lowest marks and are in need of maintenance and upgrades in the coming decade [40].

Politicians, relevant ministries (for example, the Ministry of Environment and Natural Resources, and Ministry of Tourism, Industry, and Innovation) and agencies working to increase Iceland's share of renewable energy and reduce greenhouse gas emissions may wish to take note of environmental trade-offs linked to the fossil fuel consumption of tourists, especially via cruise ships, international aviation, and rental car usage. Cruise ship tourism has also become a potentially significant source of pollution in the last few years. Cruise ships are associated with a number of negative environmental effects including air pollution, polluting discharges such as sewage, bilge oil, and chemicals, and subsequently, greenhouse gas emissions [53]. These impacts have yet to be quantified in Iceland, although cruise ship passengers have increased from about 28,000 in 2001 to about 145,000 in 2018 [54], which is an approximate increase of 420%. The hiring of rental cars has increased considerably in the last few years from around 5000 rental cars in 2006 to 21,000 in 2016 [47], and they now form almost 10% of the car fleet in Iceland [48]. Apart from the pressures on infrastructure, the increase in cars can lead to more traffic congestion and air pollution [49], and greenhouse gas emissions [50], especially in the busy capital region. The transportation sector has already been singled out as a major target area for improvement in regard to increasing the sustainability of tourism in Iceland [28], as well as being one of the nation's main policy avenues for climate action set out in Iceland's Climate Action Plan for 2018–2030 [51].

Concern was also voiced during the focus groups and reflected in the quantitative outcomes that some migrant workers in the Icelandic tourism industry were exploited and abused during their time working in Iceland. These concerns also been voiced in the English-language media in Iceland [61,62]. This should be of concern to various institutions in Iceland, including the Red Cross, municipalities, the police (especially in relation to stories of human trafficking) and the Ministry of Welfare.

Outcomes from this study should be of interest to a very broad array of domestic stakeholders, including individuals training to work in the Icelandic tourism sector, service providers, and policymakers who are tasked with maximising the benefits of synergies and either minimising the extent of trade-offs, or finding ways of intervening to transform these into synergies. They should also be relevant to academics specialising in tourism studies, as well as those from other disciplines seeking straightforward and practical methodologies that can be deployed to evaluate the contribution of economic sectors to performance across all the SDGs.

## 5.5. Contribution of Iceland's Tourism Sector to Meeting the SDGs

It was made clear to focus group participants that they were asked to assess the contribution of tourism to meeting or not meeting the SDGs and their respective targets. They were specifically requested not to evaluate whether a particular SDG or target was being met. However, it is important to consider the outcomes from this study in the light of Iceland's current performance across the SDGs.

A recent evaluation by the OECD reviewed the SDG performance for all the member states. In the case of Iceland, it was found that the nation had already achieved 17 of the targets based on the data available for 111 of the 169 targets [63]. The nation was compliant in areas relating to adult information and communication skills, air quality, and the share of renewable energy. Even though Iceland was compliant, outcomes from this study suggest that the tourism industry presents one of the few drawbacks linked to even better performance for air quality and the share of renewable energy in Iceland. This is reflected in the transition to electric car usage being one of the main policy ambitions of Iceland's Climate Action Plan for 2018–2030 [51]. Equally, objectives 12, 13, and 14 of Iceland's Climate

Action Plan recognise the environmental impacts of cruise ships and shipping, seeking to increase clean energy use for ferries, increasing the share of renewable energy utilised by ships, and advancing electrical infrastructure in harbors, respectively [51].

The OECD assessment also observed several challenges for Iceland in meeting the SDGs, with the nation considered to be very far away from meeting 5% of the targets [63]. These include targets relating to energy intensity and hazardous waste. The outcomes from this study suggest that the Icelandic tourism industry is unlikely to make either a positive or negative contribution to meeting the targets related to energy intensity or hazardous waste.

Iceland was assessed as being furthest away from meeting the SDGs on energy, sustainable production, and biodiversity (SDGs 7, 12, and 15, respectively) [63]. There are parallels with the results of the focus groups from this study. Their assessment revealed two trade-offs linked to SDG7 and two trade-offs for SDG15. Trade-offs linked to SDG7 concerned potential conflicts between increased renewable energy generation and the need to preserve nature for the benefit of tourists. This argument is part of an ongoing debate in Iceland about whether to establish a national park in the central highlands of Iceland, which would preserve the landscapes for Icelanders and tourists [64]. Although forest-based tourism is very limited in Iceland, focus group participants also recognised the tendency to plant non-native tree species as part of Iceland's programme of afforestation, which is a strategy that is mainly aimed at sequestering greenhouse gas emissions in pursuit of Iceland's climate change objectives. This approach was deemed to be contrary to the biodiversity objectives of SDG15.

### 5.6. Broader Applicability of Methods to Other Contexts

The methodological approach adopted in this paper has relevance and applicability to other studies seeking to acquire a conceptual understanding of the links between a specific sector of an economy and its contribution to SDG outcomes. The study outcomes may also be of particular interest to other nations who rely heavily on nature-based tourism, such as New Zealand, Australia, and Costa Rica. Equally, the outcomes pertaining to developing nations with significant tourism sectors may be very different. Nature-based tourism has long been advanced as a means of generating economic growth, particularly in least economically developed African states [65]. If a similar study to this one were to be adopted in a developing nation, the results might be quite different. This study found no synergies or trade-offs relating to 126 of the 169 targets (74.6%) across the 17 SDGs. Very often, this was because of the manner in which the targets were worded, which rendered objectives specific to developing nations or small island states. Due to the lack of flexibility to encompass separate objectives for developed nations, such as Iceland, many of the targets were deemed by the focus group participants to be irrelevant, especially in the social and institutional sessions. Many more of these targets would very likely be relevant and synergistic with tourism and the sector's contribution to wealth creation in developing nations, for instance those relating to poverty eradication, access to basic services, and ensuring the full and active participation of women in employment.

### 5.7. Methodological Limitations

Insights gleaned from focus groups rely heavily on the availability and willingness of experts to contribute to the panels. Although the researchers made an exhaustive effort to identify and source experts who were best suited to contribute to the deliberations, a small number were unavailable—for example, a representative from the police for the institutionally themed session—and some cancelled their participation on the day. This may have had an impact on the results in ways that are difficult to quantify. Equally, the irrelevance of many of the SDG targets to a developed nation such as Iceland, or a persistent failure to identify links between the Icelandic tourism sector and the SDG targets, may have led to some experts becoming frustrated with the evaluative process.

The scoresheet system was a useful means of establishing the conceptual links between the Icelandic tourism sector and the SDGs, but the extent of the identified synergies and trade-offs should be considered with some degree of caution. Furthermore, the arbitrary decision on the part of the

researchers to classify all the mean target outcomes in the range of −1 to +1 as neither synergies nor trade-offs may mean that some minor synergies and trade-offs were overlooked. This study does not provide a substitute for quantitative evaluations of impacts, but, especially in the case of trade-offs, rather implies areas needing further evaluation, monitoring, and consideration by the Icelandic Tourism Task Force, which is focused closely on the local sustainability impacts of Icelandic tourism. Additionally, the extent of trade-offs and synergies identified in this study may in part be reflective of emotional responses to the issues involved, for instance, the extent of social impacts relating to human trafficking. That is not to say that this impact is minor in actuality, but rather that its extent needs further evaluation.

Alternative deliberative techniques, such as the Delphi method, were rejected as the selected methodology for this paper due to the forging of a consensus rather than the elicitation of multiple values and viewpoints on the hotly debated topic of the Icelandic tourism sector. However, the adoption of the Delphi method would also have offered some advantages, in particular through the provision of a greater range of statistical information and avoidance of biases generated by outlier information.

## 6. Conclusions

The complex interactions between the SDGs and their respective targets have demanded further analysis of the links between key economic sectors and performance outcomes across all of the SDGs' 169 targets. This study used four theme-based focus groups and evaluative scoresheets to determine the synergies and trade-offs pertaining to Iceland's tourism sector, which has almost singlehandedly been responsible for transforming the nation's economy following its financial crisis of 2008. Based on the results, it was determined that there were a total of 32 synergies and 11 trade-offs across the SDGs' 169 targets. Key areas for Icelandic policymakers to focus on in the next few years include reducing greenhouse gas emissions associated with the transportation of tourists to and within Iceland, particularly via aviation, cruise ships, and rental cars. Equally, attention needs to be paid to the pressing demands on local infrastructure stimulated by the influx of tourists to Iceland, particularly the nation's road network and sewage systems. From a tourism management perspective, maximising synergies across the SDGs' economic dimensions will require the retention of a considerable volume of tourists to Iceland which equates to, many multiples greater than the scale of the national population. Mitigating trade-offs will necessitate policy interventions by various governance institutes and sound investment to minimise the negative environmental impacts of Icelandic tourism and ensure that critical infrastructure is sufficient in scale and standard.

This study stimulates several ideas for further research. In particular, greater consideration needs to be given to the particular policy initiatives that could be applied to minimise the extent of trade-offs and opportunities to transform these into synergies. Additionally, the contributions of local Icelandic communities, which are heavily dependent on tourism, need to be considered in more detail linked to Iceland's SDG performance. The methodology adopted in this paper could also be applied to other key sectors of the Icelandic economy, such as fisheries, in order to gain a broader portrayal of the relationship between economic sectors and SDG performance. All of these research lines are equally relevant to other nation-states.

**Supplementary Materials:** The following are available online at http://www.mdpi.com/2071-1050/11/15/4223/s1. Table S1: Evaluative scoresheets, including all SDGs and respective targets.

**Author Contributions:** Conceptualization, D.C.; Formal analysis, D.C.; Investigation, D.C. and N.S.; Methodology, D.C. and N.S.; Project administration, D.C. and N.S.; Supervision, B.D., L.J. and S.Ó.; Validation, D.C.; Writing—original draft, D.C. and N.S.; Writing—review & editing, D.C., B.D., L.J. and S.Ó.

**Funding:** This research was funded by NordForsk (grant number 76654) via their financial support to the Nordic Centre of Excellence ARCPATH (Arctic Climate Predictions—Pathways to Resilient, Sustainable Communities).

**Conflicts of Interest:** The authors declare no conflict of interest. The funders had no role in the design of the study; in the collection, analyses, or interpretation of data; in the writing of the manuscript, or in the decision to publish the results.

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
