# Peer review of "Synergies and Trade-Offs in the Sustainable Development Goals—The Implications of the Icelandic Tourism Sector"

_sustainability, doi:10.3390/su11154223_

Round 1

Reviewer 1 Report

The article presents a very interesting case study of synergies and trade –offs between the national tourism sector in Iceland in the context of 17 goals defined within the framework of the UN Agenda “Transforming our World”.  The article has been correctly constructed and the conclusions are presented in a transparent and comprehensive way.

After reading the article I would like to make the following comments and suggestions:

·         methodology in Section 4 – (1) what was the initial number of experts? (2) I have doubts about deleting the interviews, at this point their verification or re-analysis is not possible. Could you provide any deeper explanation? (3)  To eliminate the absence risk of several experts (with such a small group every absence is noticeable), it is worth considering to plan a mixed system supported by an individual interview (or two sessions with different participants in each of the theme-based focus groups).  My understanding is that the missing information (from the absent experts) have been included in some way by the authors of the study and supplemented in Section 5.4. and the following ones. (4) What was the structure of participants (business, academia, NGOs, tourism organizations and governmental institutions - what about the local government experts?)?

·         presentation of the results in Section 5: it is worth considering the presentation in e.g.  graphic form , which of  17 SDG were indicated by the experts with the domination of synergies or trade-offs (… 32 synergies and 11 trade-offs were identified…).

·         missing parts 5.7.4 Institutional impacts.

Author Response

Reviewer 1

The article presents a very interesting case study of synergies and trade –offs between the national tourism sector in Iceland in the context of 17 goals defined within the framework of the UN Agenda “Transforming our World”.  The article has been correctly constructed and the conclusions are presented in a transparent and comprehensive way.

Thank you for your positive comments about our study and help in further improving its content. We have responded to each of your points in turn and hope that these satisfy your concerns, especially with regards to the methodological clarifications. 

Methodology in Section 4:

(1) what was the initial number of experts?

(2) I have doubts about deleting the interviews, at this point their verification or re-analysis is not possible. Could you provide any deeper explanation?

3)  To eliminate the absence risk of several experts (with such a small group every absence is noticeable), it is worth considering to plan a mixed system supported by an individual interview (or two sessions with different participants in each of the theme-based focus groups).  My understanding is that the missing information (from the absent experts) have been included in some way by the authors of the study and supplemented in Section 5.4. and the following ones.

(4) What was the structure of participants (business, academia, NGOs, tourism organizations and governmental institutions - what about the local government experts?)?

(1) This was 53 and has been added to the text.

(2) This was a slight typo and has been corrected. It should have said “will be” deleted at the end of the research project rather than “was”. As we have stated earlier in the methods section, transcriptions and summaries of the sessions have already been written up.

(3)  Thank you for your observations. We did not carry out any individual interviews, but certainly we ensured a comprehensiveness in our approach, which we address through the broad implications of our study in section 5.4. This was based on the expert insight of the authors, all of whom are familiar with the Icelandic tourism sector and have had publications relating to the nation’s sustainability.

(4) Thank you, we have now explained this breakdown in section 4.2, as follows: The structure of participants was as follows: business (4), academia (5), NGOs (3), tourism organizations (4), and governance institutions (4). With regards to the participants attending from governance institutions, there were 3 attendees from national institutes and ministries, and 1 from the local municipal government in Reykjavik.”

Presentation of the results in Section 5: it is worth considering the presentation in e.g.  graphic form, which of  17 SDG were indicated by the experts with the domination of synergies or trade-offs (… 32 synergies and 11 trade-offs were identified…).

We agree, but contend that this is already addressed and shown clearly through the traffic-lights chart in the Appendix. We would ideally like to include this in the main body of the text, but the template for this paper does not enable the inclusion of charts/tables/diagrams in landscape format.

Missing parts 5.7.4 Institutional impacts.

Please note that it was the opinion of reviewer 2 that the article was overlong and that much of this was caused by the material in section 5.7. This has now been omitted from the revised submission, meaning your comment here is no longer applicable.

Reviewer 2 Report

The article discusses the issue of balance between the potential benefits and costs of the Icelandic tourism sector development. My overall rating is positive. However, the article is too long and contains unnecessary repetitions. My reservations have aroused by sections: 5.7 - which may be omitted without losing the information value of the article and 6 (Conclusions), which should contain recommendations related to the tourist management sphere. My anxiety concerns the selection criteria for the focus group composition “to ensure gender balance”. From a scientific point of view, only substantive competences count.

Author Response

Reviewer 2

The article discusses the issue of balance between the potential benefits and costs of the Icelandic tourism sector development. My overall rating is positive. However, the article is too long and contains unnecessary repetitions. My reservations have aroused by sections: 5.7 - which may be omitted without losing the information value of the article and 6 (Conclusions), which should contain recommendations related to the tourist management sphere. My anxiety concerns the selection criteria for the focus group composition “to ensure gender balance”. From a scientific point of view, only substantive competences count.

Thank you for your overall positive assessment of our work. We agree with your constructive comments concerning its improvement and have responded as follows:

·         We agree with your observations about Section 5.7. This has been removed from the paper. All related references (no 66-89 in first submission) have also been removed. This makes the paper more concise and focused.

·         Conclusions – we agree that specific recommendations for tourism management were missing from the first submission. These are measures that maximise the economic synergies of the expanded Icelandic tourism sector and minimise trade-offs, and have been added and highlighted.

·         Focus group composition – we may have communicated this poorly, as ensuring gender balance was not the main goal of our selection process but was considered when forming the initial pool of 53 potential participants. As you will note from the text underneath point c in section 4.2, of the eventual participants (n = 20), 60% were female, so no effort was made at this point to ensure gender balance but rather we emphasised representativeness in terms of expert participation. You will also note the text we have added in response to reviewer 1 in relation to the structure of focus group attendees i.e. whether they were employed by a business, NGO, academic or governance institute. Again, this reaffirms that our efforts were focused on an even spread of expert participants, rather than any sociodemographic specifics, as we considered this to be more important.

Reviewer 3 Report

This review primary concern is to outline the strengths and weaknesses of the paper entitled “synergies and trade-off of the sustainable development goals: the implications of the Iceland tourism sector” and, in turn, to give some pointers to the authors. On the one hand, the paper successes to pose an interesting subject matter and gives credit to the task of dealing with this topic. Similarly, the paper is well structured in that its sections are clearly addressed. Likewise, the paper comes up with a clear classification and the authors perform a discussion that might be of interest from a practitioner point of view. On the other hand, the authors do not provide the readers with the core definitions of the paper, the research techniques do not seem the most suitable for achieving the objectives and I would rather the authors explained further the main features of the measuring instruments.

However, let me be more specific by making specific recommendations as follows:

Although everyone might be aware of the meaning of synergy and trade-off, one might be confused about the exact interpretation of both concepts in this context. As far as synergy and trade-off are referring to key variables in this paper, the authors should define these concepts by quoting scientific sources. Please, define, quote and specify the exact meanings of these concepts.

SDGs is another key variable in this paper but the authors do not pinpoint systematically what specific aims encompass the sustainable development goals. Hence, let me suggest that the authors provide more information about these goals by building a table and explaining each of these goals.

The first part of the literature review is quite systematic in that the authors explain a set of works one by one. Nevertheless, in the section dedicated to explaining the sustainability impacts on Iceland, the authors build table 1 (line 200) to lay out a list of dimensions whose assignation into synergies and trade-off categories they do not clarify. What do you mean by “adapted from the framework”? What do you mean by “informed”? I think listing the dimensions is arguably as important a task as explaining how the authors “adapted” and “informed” this classification. Please, point out how you adapted and how you gather the information to be informed by pointing out the criteria you considered to get this classification.

The authors state that they used a focus group interview technique but they do not justify why this specific technique is the most suitable for this research. There is no doubt that this technique is especially suitable for gathering information about opinions and values but, I am afraid, this technique is not the most feasible if we deal with expert opinions. In fact, there is every likelihood that the Delphi research technique had been more appropriate. Not only had a Delphi been more suitable for gathering expert opinions but also keeping the anonymity and calculating a better range of statistical information. It goes without saying that the Delphi technique easily achieves a desirable level of consensus between the experts’ opinions and avoids biases derived from the outliers opinions. Why did you not use a Delphi instead? Why did you choose a focus group interview? I am afraid a focus group is especially useful if the researcher want to measure the social influence and how the focus group members interact one another.

It is hard to fault how the authors describe the use of the focus group application, but nothing they say about the use of the stakeholder analysis (line 231). How did you employ this technique? Did you distinguish different phases by identifying the stakeholders, prioritising the stakeholders with power and interest criteria and, finally, understanding your key stakeholders? Please, explain how you apply this technique.

What do you mean by hypotheses? I am sorry but I do not see them in this paper (line 281).

Did you use a questionnaire to gather information from the experts? The authors state that they designed a type of scale with 7 points in order to measure trade-off and synergies. Nevertheless, let me suggest the authors provide more information about the question they raised.

I hope these comments might be of help in improving the paper and encourage the authors to move forward.

Author Response

Reviewer 3

This review primary concern is to outline the strengths and weaknesses of the paper entitled “synergies and trade-off of the sustainable development goals: the implications of the Iceland tourism sector” and, in turn, to give some pointers to the authors. On the one hand, the paper successes to pose an interesting subject matter and gives credit to the task of dealing with this topic. Similarly, the paper is well structured in that its sections are clearly addressed. Likewise, the paper comes up with a clear classification and the authors perform a discussion that might be of interest from a practitioner point of view. On the other hand, the authors do not provide the readers with the core definitions of the paper, the research techniques do not seem the most suitable for achieving the objectives and I would rather the authors explained further the main features of the measuring instruments. However, let me be more specific by making specific recommendations as follows:

Thank you for your detailed appraisal of our submission and constructive comments for its improvement. We have responded to each of your comments in turn.

Although everyone might be aware of the meaning of synergy and trade-off, one might be confused about the exact interpretation of both concepts in this context. As far as synergy and trade-off are referring to key variables in this paper, the authors should define these concepts by quoting scientific sources. Please, define, quote and specify the exact meanings of these concepts.

We agree that most readers will possess understanding of these widely-used terms, however, we still think this is a good point and worthy of a brief explanation in the paper. We have therefore added a footnote in the introduction to the paper. With regards to definitions, we have taken these from the paper by Singh et al. (2008), a study which we further discuss in Section 2. The added text is as follows: In this study, we define synergies and trade-offs in accordance with the study by Singh et al. (2018) [7]. Synergies are understood to be co-benefits, occurring in alignment with the various activities of the Icelandic tourism sector. Trade-offs are hindrances and drawbacks linked to the same set of activities.“

SDGs is another key variable in this paper but the authors do not pinpoint systematically what specific aims encompass the sustainable development goals. Hence, let me suggest that the authors provide more information about these goals by building a table and explaining each of these goals.

This is a good point and we have included a new column in Table 2 (Section 4.1), which provides brief details about the aims of each SDG.

The first part of the literature review is quite systematic in that the authors explain a set of works one by one. Nevertheless, in the section dedicated to explaining the sustainability impacts on Iceland, the authors build table 1 (line 200) to lay out a list of dimensions whose assignation into synergies and trade-off categories they do not clarify. What do you mean by “adapted from the framework”? What do you mean by “informed”? I think listing the dimensions is arguably as important a task as explaining how the authors “adapted” and “informed” this classification. Please, point out how you adapted and how you gather the information to be informed by pointing out the criteria you considered to get this classification.

The paper states “(adapted from the framework of [29] and informed by [27] and [18])”. Reference 29 refers to the framework of Hall (2008), which delineated three main dimensions of tourism-related synergies and trade-offs: economic, environmental and socio-cultural. Hall also delineated two sub-dimensions in relation to the economic and socio-cultural dimensions. We have maintained adherence to this framework. However, when it comes to the synergies and trade-offs specific to the Icelandic tourism sector, we have relied on references 18 and 27, and in this sense Table 1 is “informed” by Icelandic-specific information. In particular, reference 27 constitutes a very recent book chapter by Jóhannsdóttir et al. (2019), which addressed these issues. As the text notes, Table 1 is a summary of the output contained in this chapter, structured in accordance with the framework of Hall (2008): “A recent book chapter by [27] and paper by [18] outlined the various economic, environmental and social sustainability impacts of Iceland’s expanded tourism sector. In this section, the aim is not to repeat the level of detail contained in a very recent publication, but rather to provide a succinct summary of the synergies and trade-offs described in its contents.” For ease of reference, this text in Section 3 has been highlighted in yellow, although it is unchanged from the first submission, as we feel it offers a sufficient explanation and the other reviewers have been satisfied. However, we have adjusted the description text in lines 208 and 209 to state more clearly that the table is structured in accordance with the framework of Hall (2008).

The authors state that they used a focus group interview technique but they do not justify why this specific technique is the most suitable for this research. There is no doubt that this technique is especially suitable for gathering information about opinions and values but, I am afraid, this technique is not the most feasible if we deal with expert opinions. In fact, there is every likelihood that the Delphi research technique had been more appropriate. Not only had a Delphi been more suitable for gathering expert opinions but also keeping the anonymity and calculating a better range of statistical information. It goes without saying that the Delphi technique easily achieves a desirable level of consensus between the experts’ opinions and avoids biases derived from the outliers opinions. Why did you not use a Delphi instead? Why did you choose a focus group interview? I am afraid a focus group is especially useful if the researcher want to measure the social influence and how the focus group members interact one another.

As the reviewer notes, one of the main aims of our work was to elicit opinions and values from a representative spectrum of experts in small focus groups. With regards to feasibility, we agree that the Delphi method would have offered some advantages in terms of developing expert opinions and gathering greater statistical information. However, the Delphi method is used to systematically combine expert opinion in order to arrive at an informed group consensus, often being applied as a forecasting tool. In contrast, the aim of our study was not to form a consensus from experts, but rather to ensure that the full breadth of expert opinion and perspectives were elicited. In some ways, we wanted the opposite of consensus, seeking to observe differences of opinion by representative experts across different stakeholder groups. Furthermore, we wanted to see when and why they disagreed in order to better understand each representative’s perspective – this is much more easily done when the aim of the deliberative exercise is not to arrive at a single agreed outcome. Moreover, the scoresheets were completed independently and participants were reminded at the start of each focus group session that the aim was not necessarily for them to select the same numeric values on the seven-point scale as other attendees.

We agree, though, that the paper need to make a better justification of its choice of focus groups over alternatives. We have therefore added text and citations in Section 4.1, explaining why we opted for focus groups over the Delphi method, and voiced in Section 5.7 of the discussion that an alternative approach could have reaped a greater range of statistical information for analysis.  

It is hard to fault how the authors describe the use of the focus group application, but nothing they say about the use of the stakeholder analysis (line 231). How did you employ this technique? Did you distinguish different phases by identifying the stakeholders, prioritising the stakeholders with power and interest criteria and, finally, understanding your key stakeholders? Please, explain how you apply this technique.

Thank you for your commendation of our description of the methodology for the focus groups. With regards to stakeholders, we strove to ensure that there was no duplication in terms of expert input and that there was an even split in terms of background. It should be noted that Iceland is a very small country (in population) with a limited number of key stakeholders specific to the tourism industry. Key stakeholders were identified by the expert team (the authors) during an internal review of the recently produced stakeholder map of the Icelandic tourism sector (Nordic Council of Ministers, 2019, see citation no. 33).  One representative from all key stakeholders in the five stakeholder categories (business, academia, NGOs, tourism organizations, and governance institutions) attended the focus group sessions. This approach has been clarified slightly in section 4.2.

What do you mean by hypotheses? I am sorry but I do not see them in this paper (line 281).

Apologies, this was text that was included from the initial download of the template for this paper, and was never meant to be included in the submission. It has now been deleted.

Did you use a questionnaire to gather information from the experts? The authors state that they designed a type of scale with 7 points in order to measure trade-off and synergies. Nevertheless, let me suggest the authors provide more information about the question they raised.

We designed a scoresheet and this is shown in Appendix A. The numeric outcomes of the process are reported in Appendix B, which demonstrates the traffic-lights methodology described in Section 4 of the paper.

I hope these comments might be of help in improving the paper and encourage the authors to move forward.

Thank you again for your constructive assistance and time taken to review our paper.

Round 2

Reviewer 1 Report

I find all your replies and comments satisfying. I'm waiting impatiently for the final on-line version.

Reviewer 2 Report

I am satisfied with the authors` respond to the review comments. 

Reviewer 3 Report

Dear authors,

it is hard to fault how specifically and clear you have given an answer to all my concerns and comments. In my opinion, the paper is bringing into focus more profesional and pragmatic results than new scientific knowledge and, thus, it might be of great interest from a practitioneer point of view. Hence, it is my belief that not only is the paper of interest for a wide audience but also free of any serious shortcomings. In fact, the applications of methods seem quite rigorous.

Therefore, there is no doubt that the paper describes a interesting issue and follows correct procedures. On this basis, there is no objection to recommending we publish it.

Regards